# The First Optimization Process from Cultivation to Flavonoid-Rich Extract from *Moringa oleifera* Lam. Leaves in Brazil

**DOI:** 10.3390/foods11101452

**Published:** 2022-05-17

**Authors:** Larissa Marina Pereira Silva, Maria Raquel Cavalcanti Inácio, Gualter Guenter Costa da Silva, Jucier Magson de Souza e Silva, Jefferson Romáryo Duarte da Luz, Maria das Graças Almeida, Edgar Perin Moraes, Debora Esposito, Leandro De Santis Ferreira, Silvana Maria Zucolotto

**Affiliations:** 1Laboratory of Pharmacognosy, Research Group of Bioactive Natural Products (PNBio), Postgraduate Program in Drug Development and Technological Innovation, Federal University of Rio Grande do Norte, Natal 59012-570, Brazil; larissamarinaps@gmail.com (L.M.P.S.); quelquimica@yahoo.com.br (M.R.C.I.); 2Plants for Human Health Institute, North Carolina State University, 600 Laureate Way, Kannapolis, NC 28081, USA; daesposi@ncsu.edu; 3Laboratory of Quality Control of Medicines (LCQMed), Department of Pharmacy, Federal University of Rio Grande do Norte, Natal 59012-570, Brazil; lean_sf@yahoo.com.br; 4Agricultural Sciences Unit, Federal University of Rio Grande do Norte, Macaíba 59280-000, Brazil; gualtermve@gmail.com (G.G.C.d.S.); jucier.magson@gmail.com (J.M.d.S.e.S.); 5Multidisciplinary Research Laboratory (LabMult), Department of Clinical and Toxicological Analyzes, Federal University of Rio Grande do Norte, Natal 59012-570, Brazil; jeffduarte00@gmail.com (J.R.D.d.L.); mgalmeida84@gmail.com (M.d.G.A.); 6Organic Chemistry and Biochemistry Laboratory, Amapá State University (UEAP), Av. Presidente Vargas, s/n, Centro, Macapá 68900-070, Brazil; 7Chemometrics and Biological Chemistry Group (CBC), Institute of Chemistry, Federal University of Rio Grande do Norte, Natal 59078-970, Brazil; edgarmoraesufrn@gmail.com; 8Department of Animal Science, NC State University, 120 Broughton Drive, Raleigh, NC 27695, USA

**Keywords:** flavonoids, *Moringa oleifera*, response surface methodology, ultrasound, cultivation, HPLC-ESI-QTRAP-MS/MS

## Abstract

Flavonoids are significant antioxidant and anti-inflammatory agents and have multiple potential health applications. *Moringa oleifera* is globally recognized for its nutritional and pharmacological properties, correlated to the high flavonoid content in its leaves. However, the bioactive compounds found in plants may vary according to the cultivation, origin, season, and extraction process used, making it difficult to extract reliable raw material. Hence, this study aimed to standardize the best cultivation and harvest season in Brazil and the best extraction process conditions to obtain a flavonoid-rich extract from *M. oleifera* as a final product. Firstly, ultrasound-assisted extraction (UAE) was optimized to reach the highest flavonoid content by three-level factorial planning and response surface methodology (RSM). The optimal cultivation condition was mineral soil fertilizer in the drought season, and the optimized extraction was with 80% ethanol and 13.4 min of extraction time. The flavonoid-rich extract was safe and significantly decreased reactive oxygen species (ROS) and nitric oxide (NO) in LPS-treated RAW 264.7 cells. Lastly, the major flavonoids characterized by HPLC-ESI-QTRAP-MS/MS were compounds derived from apigenin, quercetin, and kaempferol glycosides. The results confirmed that it was possible to standardize the flavonoid-rich extract leading to a standardized and reliable raw material extracted from *M. oleifera* leaves.

## 1. Introduction

The application of medicinal plants as health-promoting food additives or medicinal products has been exponentially expanding because of their nutritional and pharmacological properties [1]. This is associated with the presence of health-relevant secondary metabolites [1,2]. A consistent correlation between the intake of rich sources of natural antioxidants (i.e., plant phenolics including flavonoids) has been well documented in recent years to prevent or mitigate various chronic inflammatory diseases [3,4,5,6]. 

*Moringa oleifera* Lam. (order Brassicales, family Moringaceae) is a tree native to South Asia that is currently cultivated in numerous tropical and subtropical regions worldwide for multiple uses [1,2,7]. It is a winter-hardy and drought-resistant plant, found in arid and semiarid areas [8], such as the Caatinga biome of Northeastern Brazil. *M. oleifera* is commonly known as “moringa”, “drumstick tree”, “horseradish tree”, “tree of life”, or “miracle tree” [1]. It is characterized as a fast-growing perennial tree with easily propagated winged seeds [1,9,10,11]. This plant is considered one of the most valuable trees in the world since almost all of its parts are edible, and their leaves can be widely consumed in different preparations (e.g., fresh, in a salad, cooked, or as a dried powder) as healthy-relevant food [2,7]. Preventing specific inflammatory processes associated with cellular stress, antioxidant, anti-inflammatory, and anti-diabetic properties has primarily been attributed to the high flavonoid content in its leaves [7,11]. Its leaves are relevant to treating malnutrition since they contain 22–37% proteins by dry weight, vitamins A and C, minerals (iron, calcium, potassium, among others), carotenoids, and all the essential amino acids [7,10,11]. It is significant to highlight that the leaves contain some anti-nutritional factors [2] with no toxicity in vitro and in vivo [12,13].

The use of *M. oleifera* leaves has recently increased in many developing countries, known to practice traditional herbal medicine for the primary population’s health needs. In this context, different preparations of *M. oleifera* leaves (raw dried powder or capsules) have been manufactured and sold in different ways worldwide [11,14]. Unfortunately, its consumption increased with the exponential growth of misinformation and irrational use in Brazil. Moreover, some products were sold with therapeutic effect claims on supplement product labels, which is illegal in Brazil [15]. Considering this incorrect usage and lack of supporting data, the National Health Surveillance Agency (ANVISA) published Resolution No 1.478 on 3 June 2019, prohibiting the use and commercialization of Moringa-based products in Brazil until scientific evidence related to its safety is provided [16].

Studies dealing with *M. oleifera* leaves have shown that the bioactive compounds may vary depending on the cultivation condition, geographical origin, harvesting season, edaphoclimatic factors, and the utilized extraction process [17,18]. These variables make it difficult to have standardized and reliable raw materials [18]. Compared with traditional techniques (i.e., maceration, Soxhlet), UAE may improve the extraction yield of total phenols and flavonoids in the leaves [4,19,20,21,22]. UAE is becoming the most frequently used method for industrial application because it is upscalable. It uses a lower extraction temperature, shorter time, requires less solvent, increases mass transfer, and is compatible with green-extraction principles [18].

In this study, RSM (response surface methodology) was selected to evaluate the variables of the UAE process and to optimize the extraction of flavonoids. Ethanol was selected as the extraction solvent because it has the ability of extracting a wide polarity of compounds, it can be easily evaporated from the final product, and it is in line with the global trends in developing herbal medicine and food. RSM is an accurate and effective multi-factor method used to develop and optimize extraction [23]. It also maximizes the extraction yield of targeted compounds in natural products with a minimal number of experiments [23]. During the COVID-19 pandemic, the world woke up to a serious problem related to the supply dependence of many countries. Brazil has the greatest biodiversity in the world, however it imports raw materials from other countries to produce pharmaceutical products, cosmetics, or supplements. In this sense, our work can contribute to strengthen the development of all productive chains in the country, from cultivation to the finished product.

Thus, the optimized process was applied to *M. oleifera* leaves from different soil cultivations and harvest seasons, and the best condition was chosen based on the yield of the total of phenols, flavonoids, and proteins. The fingerprint of the best extract by HPLC-ESI-QTRAP-MS/MS enabled the characterization of nine compounds, mainly glycosidic derived from apigenin, quercetin, and kaempferol. Finally, the cell viability, the suppression of nitric oxide generation, and reactive oxygen species in LPS-treated RAW 264.7 cells were evaluated. 

Many studies refer to the use of standardized raw material, which usually implies only a chemical standardization [24]. In this sense, our proposal is broader and aims to standardize all stages of the productive chain to develop a flavonoid-rich extract from *Moringa* leaves, from cultivation to a dried extract to strengthen the national supply chains Therefore, this study established the best cultivation conditions and harvest season through monitoring flavonoid content in dried extract in order to achieve these goals. This process can be applied further in the food or pharmaceutical industries to develop new supplements or medicines that contain a flavonoid-rich extract from *Moringa oleifera* to be used as a coadjuvant to treat chronic inflammatory diseases.

## 2. Materials and Methods

### 2.1. Plant Material

#### 2.1.1. Cultivation Conditions

This experiment was conducted in open field cultivation at Escola Agrícola de Jundiaí (Macaíba city, Rio Grande do Norte state, Brazil), (GPS coordinates: lat: −5.900133 and long: −35.357028), with no plant management (e.g., irrigation, pruning, weeding, or pest control) to simulate natural environmental conditions. A voucher was identified by the botanist Anderson Fontes and deposited in the Herbarium of the Center of Biosciences of the Federal University of Rio Grande do Norte, Brazil (UFRN 25423). The research was authorized by the National System for the Management of Genetic Heritage and Associated Traditional Knowledge (SISGEN process No. A5DB251).

The seedlings were provided by Raros Naturals (Macaíba, Brazil) and were replanted in the experimental field at 45 days of age. The fertilization was applied in a 5 cm layer above the soil in June 2018, except for the control group. The randomized block design was composed of 12 individual plants (3 × 3 m distance between plants and margins). The experiment was conducted according to the following conditions: 100% mineral soil fertilizer (131 kg/ha of the combination of ammonium sulfate ((NH_4_)_2_SO_4_, Usifértil^®^), triple superphosphate (P_2_O_5_, Fertine^®^), potassium chloride (KCl, Yara^®^)) and 50% mineral with 50% organic soil fertilizer (composting (7.0 t/ha) or Biochar (3.0 t/ha)).

Four experimental groups were evaluated: Control (untreated local soil, denominated control soil cultivation); MIN (100% mineral soil fertilizer); MIN-C (50% mineral soil fertilizer with 50% composting); and MIN-B (50% mineral soil fertilizer with 50% Biochar).

#### 2.1.2. Sample Harvests and Preparation

Fresh *M. oleifera* leaves were harvested from 12 individual plants for each soil cultivation condition. The harvest was well-distributed to homogenously collect leaves of different parts of each tree. Harvests were performed on rainless mornings and in two different seasons to analyze the seasonality effect on the content of specialized metabolites (such as flavonoids). The average rainfall during the first harvest was 6.1 mm/month in August 2018 (drought season), and 206.3 mm/month in the second harvest February 2019 (rainy season) [24]. The leaves of each individual were stored separately in Ziploc^®^ bags with liquid nitrogen immediately after harvest. Then, they were crushed with liquid nitrogen in a well-distributed manner to stop phytochemistry metabolism (*quenching*). Each cultivation condition was analyzed as a sample pool. Each sample pool was extracted in triplicate to ensure analytical reproducibility and stored at −80 °C [25].

### 2.2. Experimental Design of Optimized Ultrasound-Assisted Extraction

The *M. oleifera* leaves were extracted with different percentages of ethanol (50%, 65%, 80%) (1:10 extraction ratio *w*/*v*). The mixture was extracted using an ultrasonic bath (Kondortech, CD-4860, power 310 W, frequency 50 kHz), and the RSM package was performed to identify the optimal parameters [23,25,26,27]. This methodology was composed of two distinct stages: modeling and displacement.

With the effect of two independent variables (2^2^: percentage of ethanol (*X*_1_), and extraction time (*X*_1_)), the total phenol and flavonoid contents were investigated using three-level factorial planning (−1%, 0%, 1%; 50%, 65%, 80%; 10, 15, 20 min). The complete optimization design was carried out in random order and consisted of 15 experimental runs, including three replicates at the central point to estimate the pure error and the adequacy of the fitted model. The experimental data were analyzed by multiple linear regression (MLR), first-order (FO), and second-order (SO) models. The variability was calculated by the coefficient of determination (*R*^2^) using the R statistical software program [28] with the quality Tools package version 1.55 [29].

Each extract was vortexed for 1 min and centrifuged at 3000× *g* for 30 min (4 °C). The supernatant (1.5 mL) was recovered, transferred to a pre-weighed Eppendorf^®^ tube, and dried in a speed vacuum concentrator followed by a freeze dryer. Each condition was carried out in triplicate and kept at 4 °C before the analyses. Therefore, the standardized extraction protocol employs an ultrasonic bath with controlled conditions, as described in Figure 1.

### 2.3. Determination of Total Phenol Content (TPC)

The TPC was determined by a Folin-Ciocalteu assay, modified by Domínguez et al. [30]. First, 25 μL of each extract (2000 μg/mL) was mixed with 125 μL of Folin-Ciocalteu reagent (diluted 10-fold) and 100 μL of sodium bicarbonate solution (7.5% *w*/*v*), making the final volume 250 μL. The blank was a mixture of 150 μL of water and 100 μL of sodium bicarbonate solution. This combination was vortexed and allowed to react for 30 min at room temperature in the dark. The absorbance was measured at 765 nm against the blank using an ELISA microplate reader (Epoch-BioTek, Winooski, VT, USA) and compared to a gallic acid calibration curve (2.5 to 100 μg/mL). The TPC was calculated as mean ± SEM (*n* = 3) and expressed as milligrams of gallic acid equivalent per g of the *M. oleifera* leaf dried extract (mg GAE/g DM).

### 2.4. Determination of Total Flavonoid Content (TFC)

The TFC was determined using the aluminum chloride colorimetric method [31]. First, 50 μL of each extract (2000 μg/mL) was mixed with 160 μL of ethanol (P.A.), 20 μL of aluminum chloride solution (1.8% *w*/*v*), and 20 μL of sodium acetate (8.2% *w*/*v*). The mixture was allowed to react for 40 min at room temperature in the dark. The absorbance was measured at 415 nm against the blank using an ELISA microplate reader and compared to a quercetin calibration curve (1.25 to 200 μg/mL). The TFC was calculated as mean ± SEM (*n* = 3) and expressed as milligrams of quercetin equivalent per g of the dried *M. oleifera* leaf extract (mg QE/g DM).

### 2.5. Determination of Total Sugars and Proteins Content

The assay of sugars was carried out using the reaction of phenol–H_2_SO_4_, with D-glucose as the standard [32]. The total protein content was estimated following the Bradford method using bovine serum albumin (BSA) as standard, with an absorbance reading performed at 595 nm [33]. The result was calculated as mean ± SEM (*n* = 3) and expressed as milligrams of glucose equivalent per g of the dried *M. oleifera* leaf extract (mg glucose/g DM), or as milligrams of BSA equivalent per g of the *M. oleifera* leaf extract (mg BSA/g DM), respectively.

### 2.6. Characterization by HPLC-ESI-QTRAP-MS/MS

The extract with the highest TPC, TFC, and total protein result was chosen to be analyzed by mass spectrometry. The analysis was performed on a Dionex Ultimate^®^ 3000 liquid chromatography (San Jose, CA, USA) consisting of a binary pump (Ultimate 3000 Pump), autosampler at 20 °C (Ultimate 3000 Autosampler), column oven (Ultimate 3000 Column Compartment), and a Diode Array Detector (Ultimate 3000 Diode Array Detector). The constituents were separated on a Phenomenex Hydro C_18_ column (150 × 4.6 mm, 2.6 μm) at a flow rate of 500 µL/min, at 35 °C. The mobile phase was a gradient mixture of 0.1% formic acid in water (A) and 0.1% formic acid in acetonitrile (B). The multi-step gradient conditions were: 1–15% of B (0–10 min), 15–20% B (10–20 min), 20–40% B (20–40 min), and 40–90% B (40–50 min), then remaining at this concentration for 5 min, after which it returned to the initial dose to equilibrate the column for the next injection.

The chromatographic system was coupled to a triple quadrupole Ion trap linear mass spectrometer (3200 QTRAP^®^ LC-MS/MS, AB Sciex, Toronto, ON, Canada, with TurboIonSpray^TM^ (AB Sciex) as an ion source, operating in the positive and negative modes. Instrument control, data acquisition, and processing were performed using Analyst^Ⓡ^ 1.5 and Chromeleon^Ⓡ^ 6.8 software via the Dionex Chromatography MS Link platform. The detection conditions applied included an electrospray ionization (ESI) Turbo Spray source operating at 600 °C with the following appropriate settings: curtain gas (nitrogen) 20 psi, ion source gas (GS1) 45 psi, ion source gas (GS2) 45 psi, collision gas (nitrogen) at medium position and ion spray voltage of 4500 V and −4500 V for positive and negative modes, respectively. Full scan data acquisition was performed ranging from 50 *m*/*z* to 400 *m*/*z* in enhanced mass spectrometry/information-dependent acquisition/enhanced product ion (MS IDA EPI) mode. The chromatographic analyses were performed at ambient temperature. The peaks were identified by comparison with an in-house mass spectral database, the literature, and open access mass-spectra databases the Metlin (https://metlin.scripps.edu, accessed on: 10 March 2022) and the MassBank (http://www.massbank.jp, accessed on: 10 March 2022).

### 2.7. Cell Culture

The following assays were carried out as previously described for Esposito et al. [34] and Grace et al. [35], using murine macrophage (RAW 264.7) cells obtained from American Type Culture Collection (ATCC, Livingstone, MT, USA) (ATCC^®^ TIB-71^TM^). Cells were routinely maintained in Dulbecco’s modified Eagle’s medium (DMEM, Life Technologies, Grand Island, NY, USA), supplemented with 100 μg/mL penicillin and 100 μg/mL streptomycin (Penstrep, Gibco, Life Technologies, Washington, DC, USA, REF#15140-122), and 10% (*v*/*v*) fetal bovine serum (FBS, Life Technologies, Long Island, NY, USA). Approximately 2.8 × 10^5^ cells/mL were kept at 37 °C and 5% CO_2_ in a humidified incubator. The *M. oleifera* leaf extract was reconstituted in 80% ethanol at 50 mg/mL stock solution, and serial dilutions of 250, 125, 50, 10, and 5 µg/mL were prepared and stored at −20 °C until later use.

### 2.8. MTT Viability Assay

RAW 264.7 cells were seeded in a sterile 96-well microplate (Nunclon^TM^ Delta Surface, Thermo Scientific, San Jose, CA, USA) with DMEM. After adhesion and confluence over 24 h, we treated them with *M. oleifera* leaf extract (MOL-Flav) serial dilutions (250, 125, 50, 10, and 5 µg/mL) in comparison with negative (80% ethanol) and positive controls (1% DMSO, dimethyl sulfoxide) and then incubated them again (37 °C, 5%, CO_2_). After 18 h, the treated macrophages were exposed to the 3-(4,5-dimethylthiazol-2-yl)-2,5-diphenyl-tetrazolium bromide (MTT) reagent for 4 h. The assay was performed in triplicate. The purple formazan crystals were measured on a microplate reader (Synergy H1, Biotek, Winooski, VT, USA) at 570 nm. 

### 2.9. In Vitro Reactive Oxygen Species (ROS) 

The cells were seeded into a sterile 24-well plate (Nunclon^TM^ Delta Surface, Thermo Scientific) with DMEM. After adhesion and confluence over 24 h, they were exposed to a fresh fluorescent medium of 50 μM solutions of dichlorodihydrofluorescein diacetate acetylester (H_2_DCFDA) in ethanol for 30 min. The medium was aspirated, and the cells were treated with 1 μL of the extract concentrations and the negative (80% ethanol) and positive controls of 10 μM ammonium pyrrolidinedithiocarbamate (PDTC) and 10 μL of lipopolysaccharide (LPS, from *Escherichia coli* 127: B8, 100 μg/mL). Then, the cultured cells were incubated for 24 h. The fluorescence of 2′,7′-dichlorofluorescein (DCF) was measured at 485 nm (excitation) and 515 nm (emission) on a microplate reader (Synergy H1, Biotech, Winooski, VT, USA) using the Gen 5^TM^ software program (Take 3 Session, Biotek, Winooski, VT, USA). The results were expressed as ROS production (%) relative to LPS induction.

### 2.10. Nitric Oxide Radical Inhibition (NO)

The NO inhibition was determined using the Griess reagent system (Promega Corporation, ref. G2930, Madison, WI, USA). The cells were seeded in a sterile 24-well plate (Nunclon^TM^ Delta Surface, Thermo Scientific) with DMEM. After adhesion and confluence over 24 h, they were treated in triplicate with the extract concentrations and negative (80% ethanol) and positive (10 μM dexamethasone, DEX) controls for 1 h. Next, the inflammatory response was induced by 10 ng/mL of LPS (*Escherichia coli* 127: B8, 1 μg/mL). After incubation at 37 °C and 5% CO_2_ for 18 h, 50 μL of cell-free supernatant was mixed with 50 μL of Griess reagent 1 (1% sulfanilamide) and incubated for 10 min in the dark at room temperature. Then, the plate was mixed for 1 min and centrifuged for 1 min. In sequence, 50 μL of Griess reagent 2 (0.1% N-(1-naphthyl) ethylenediamine dihydrochloride) was mixed for 1 min and centrifuged for 1 min. The absorbance was read at 520 nm on a microplate reader (Synergy H1, Biotek, Winooski, VT, USA). A calibration curve built with serial dilutions of sodium nitrite (1.56–100 µM, *R*^2^ = 0.990) was used to calculate the nitric oxide concentration. The results expressed NO production (%) relative to LPS induction.

### 2.11. Statistical Analysis

The results were expressed as the mean ± standard error of the mean (SEM). The RSM and the Student’s *t*-test determined the significance of the coefficients and standardized effects, and the Fisher’s *F*-test was performed using the Statistica 7.0 software program (Statsoft Inc., Oklahoma, OK, USA). The TPC, TFC, total sugar, and total proteins were statistically analyzed using a one-way analysis of variance (ANOVA) followed by Tukey’s post hoc test. Cell viability, NO, and ROS were analyzed by one-way ANOVA followed by Dunnett’s post hoc multiple comparison test. Both were done using the GraphPad Prism version 6.0 program (GraphPad Software, San Diego, CA, USA). All of the statistical analyses consider *p* < 0.05 as statistically significant.

## 3. Results

### 3.1. Influence of Extraction Time and Ethanol Percentage on Phenolic and Flavonoid Content

Table 1 shows the factorial planning and the results of the tested extraction parameters on TPC and TFC. The amount of TPC ranged from 447.63 to 679.21 mg GAE/g DM. The optimum conditions for TPC were found in experiment 6 (679.21 mg GAE/g DM). The amount of TFC ranged from 274.69 to 478.61 mg QE/g DM. The optimum conditions for TFC were in the experiment 8 (478.61 mg QE/g DM). Thus, the optimized conditions to obtain the highest TPC (maximum time, minimum percentage of ethanol) are the opposite of the TFC (minimum time, maximum percentage of ethanol).

### 3.2. Optimized Extraction Conditions

Response surface data were analyzed using multiple linear regression (MLR) and second-order (SO). The results are shown in Table 2. The MLR and SO were used to determine the relationship between the independent variables (time, percentage of ethanol) and the total phenolic and flavonoid contents. The *p* values indicate the degree of influence each variable had on TPC and TFC.

Based on the ANOVA results for TPC, which includes a coefficient of determination (*R*^2^), and *F* values for the dependent variables, the MLR model had no statistical significance (*p* < 0.4203). However, the SO model presented statistical significance (*p* < 0.003197). Moreover, both models were significant for TFC, MLR, and SO (*p* < 0.001982, *p* < 0.0000014, respectively). Variance analysis of the response surface revealed that the significance of the TFC was influenced by the percentage of ethanol, which was extremely significant (*p* < 0.0000014). The SO was the adequate statistical model for TFC, with *R*^2^ = 0.9524. 

The response surface curves for the effects of two independent variables on the ultrasound-assisted extraction (UAE) of *M. oleifera* leaves are shown in Figure 2. There is a linear increase in the TPC with increased extraction time. The best condition was with the highest time (20 min) and lower ethanol percentage (50%) (Figure 2B). The results revealed that ethanol concentration was the most influential variable. The TFC increased with the lowest time (10 min) and the highest ethanol percentage (80%) (Figure 2E).

These results demonstrate that the model may be used to predict the yield of total phenolic and flavonoid content. In this way, the flavonoid-rich extract from *M. oleifera* leaves (MOL-Flav) was obtained at a solid-to-liquid ratio of 1:10 g mL^−1^ of leaves, 13.5 min of extraction time, and 80% of ethanol as a solvent extractor. This is a short extraction time to obtain flavonoids from *M. oleifera* leaves [1].

### 3.3. Effect of Cultivation Condition and Harvest Eason 

Based on the optimized flavonoid-rich extraction protocol, extracts of the four cultivation conditions and two different harvests were prepared, and the yield, TPC, and TFC of extracts were evaluated. The yield is a characteristic complex associated with many other contributing traits or simply factors. Considering the soil conditions, MIN-C showed the highest yield, and it was in the first harvest (7.29%). The MIN-B condition was able to maintain the yield high during the first and second harvest seasons (6.40% and 6.15%, respectively), which was an expected long-term action due to using charcoal [36].

The highest yield was found in the first harvest for all soil conditions between the two harvest seasons. Thus, the results suggest that *M. oleifera* leaves in the drought season (1st harvest) had the higher yield compared to rainfall (2nd harvest) (Table 3).

The effects of different soil conditions and harvest seasons on the total phenolic, flavonoids, sugar, and protein contents are summarized in Figure 3. 

It is essential to emphasize the lack of a direct correlation between extract biomass yield (Table 3) and specific metabolites content (Figure 3). Since the highest yield was obtained in the MIN-C condition, the highest flavonoid content was obtained in the MIN condition, both in the first collection. The production of phenolic compounds and total sugars had the opposite profile in MIN-B (Figure 3A,C), suggesting that the same metabolic pathway could have been directed towards better sugar production. Therefore, the MIN conditions and on the first harvest had the highest levels of TPC (4.69 ± 1.39 mg GAE/g DM), TFC (5.35 ± 0.37 mg QE/g DM), and proteins (6.02 ± 1.00 mg BSA/g DM), with a statistically significant difference with the other groups, and adequate yield (Table 3). Thus, MIN was considered as the best cultivation condition (Figure 3).

### 3.4. Fingerprinting and Compound Characterization of M. oleifera Leaf Extract by HPLC-ESI-QTRAP-MS/MS

According to the results concerning TPC, TFC, and total proteins above, the extract from the MIN cultivation condition and the first harvest was chosen for the analyses in the following steps. The peaks were numbered according to their retention time (Figure 4). The characterization of individual major compounds was accomplished utilizing MS/MS spectra in the ESI negative ion mode and the retention time in comparison with the data from an in-house mass spectral database and open access mass-spectra databases the Metlin (https://metlin.scripps.edu, accessed on: 10 March 2022) and MassBank (http://www.massbank.jp, accessed on: 10 March 2022). Table 4 summarizes the nine main compounds found in the MOL-Flav. The most abundant compounds were flavonoid glycosides derived from apigenin, quercetin, and kaempferol.

The detailed information is described in Appendix A. Compound **1** showed a precursor ion *m*/*z* 191.02 [M–H]^−^ and fragments *m*/*z* 173.08 [M–H–H_2_O]^−^ and 126.86 [M–H–H_2_O–HCOOH]^−^ suggesting the molecular formula C_7_H_12_O_6_, related to the structure of quinic acid [37,38,39] (Appendix A).

Compound **2** showed a precursor ion *m*/*z* 611.35 in the negative mode and fragment ion *m*/*z* 371.26, suggesting the elimination of a hexoside group. Moreover, the characteristic fragment *m*/*z* 269.11 corroborates to assign the aglycone as apigenin, while the ion *m*/*z* 239.11 and the most abundant fragment ion *m*/*z* 209.12 are related to sugar fragmentation. The fragment ion *m*/*z* 167.00 suggests the elimination of an acetyl-hexosyl moiety. However, it was not possible to elucidate its structure, so it remains unidentified (Appendix A).

Compound **3** was identified as apigenin-6,8-*C*-dihexose (vicenin-2) with a precursor ion *m*/*z* 593.23 [M-H]^−^ (C_27_H_30_O_15_). The MS spectrum showed characteristic fragment ions of *C*-hexosyl flavones at [M-120]^−^ and [M-H-90]^−^, and product ions at *m*/*z* 383.29 [M-H-90-120]^−^, and *m*/*z* 353.25 [M-H-120-120]^−^, resulting from internal sugar fragmentation [40] (Appendix A).

Compounds **4** and **5** were characterized as isomers of apigenin hexoside with precursor ions *m*/*z* 431.33 [M-H]^−^ (vitexin) and 431.27 [M-H]^−^ (isovitexin), and *m*/*z* 433.39 (vitexin and isovitexin) in positive mode. The highest and characteristic fragments obtained in MS/MS analysis were *m*/*z* 311.23 [M-H-120]^−^ and *m*/*z* 313.28 [M+H-120]^+^ for vitexin (**4**) (Appendix A) and *m*/*z* 311.20 [M-H-120]^−^ and *m*/*z* 283.24 [M+H-30-120]^+^ for isovitexin (compound **5**) (Appendix A) [39].

The subclass of flavonols glycosides, including acetyl hexosides, are among the most described the flavonoids identified in *M. oleifera* leaves cultivated in Brazilian Northeast, t [38]. Two glycosylated quercetin derivatives (**6** and **7**) were identified and presented as characteristic fragments ions *m*/*z* 300 and 301 in negative mode, corresponding to the quercetin aglycone. The precursor ion *m*/*z* 463.21 [M-H]^−^ (C_12_H_20_O_12_) formed the fragment product ions *m*/*z* 300 and 301 [M-H]^−^ due to hexose moiety elimination. Compound **6** was identified as quercetin-3-*O*-hexose, showing typical fragment ions *m*/*z* 271.18 [M-H-30]^−^, *m*/*z* 179.09 (fragment ion characteristic of flavonol group), and *m*/*z* 151.98 (a classic retro Diels-Alder (RDA) fragment diagnostic ion), and has already been described for *M. oleifera* leaves [21] (Appendix A). Meanwhile, compound **7**, quercetin-3-*O*-acetyl-hexose, exhibited a precursor ion *m*/*z* 505.27 [M-H]^−^ (C_23_H_21_O_13_) with product ions *m*/*z* 301.18 and 300.19 [M-H-162-42]^−^, indicating the elimination of an acetyl-hexosyl group (Table 4) [37] (Appendix A).

Additionally, the MS/MS spectrum obtained for compounds **8** and **9** in negative ionization mode presented similar fragmentation patterns to kaempferol or luteolin, *m*/*z* 285.18 and 284.21 [M-H]^−^, since they had the same molecular formula. However, the distinction could be made by the MS/MS spectrum in the positive ionization mode, and since kaempferol had *m*/*z* 213 [M+H]^+^ as a characteristic product ion [41,42]. Therefore, it was possible to identify two kaempferol derivatives. The flavonoid at 25.97 min (**8**) showed a precursor ion *m*/*z* 447.22 [M-H]^−^ (C_21_H_20_O_11_) and fragments *m*/*z* 285.18 and 284.21, indicating the loss of a hexose moiety, characteristic of kaempferol-3-*O*-hexose [43] (Appendix A). Additionally, the compound at 28.24 min (**9**) revealed a similar fragmentation pattern to compound **8**, as the fragment *m*/*z* 285.15, but the precursor ion had *m*/*z* 489.22 [M-H]^−^ (C_23_H_22_O_11_). Hence, this compound was identified as kaempferol-3-*O*-acetyl-hexose [44] (Appendix A).

### 3.5. Cell Viability and Effect on Nitric Oxide and Intracellular Reactive Radical Species

Figure 5A shows the cell viability percentage (%) of RAW 264.7 cells in the presence of different MOL-Flav concentrations (5, 10, 50, 125, 250 µg/mL). The concentrations of 5, 10, and 50 µg/mL showed no cytotoxicity. However, the highest doses of 125 and 250 µg/mL reduced cell viability below 80% compared to the negative control (80% ethanol). Therefore, these cytotoxicity concentrations were not used for the following in vitro analyses.

PDTC (pyrrolidinedithiocarbamat ammonium) is a potent inhibitor of the nuclear factor-κB (NF-κB) signaling pathway, since downregulation of this pathway is essential to attenuate chronic inflammation processes in inflammatory diseases [45]. So, the effects of MOL-Flav were analyzed in comparison with PDTC. The MOL-Flav suppressed the generation of ROS in LPS-treated murine macrophage RAW 264.7 cells, between 5 and 50 µg/mL (*p* < 0.05). MOL-Flav at 10 and 50 µg/mL inhibited the ROS by around 60% compared to LPS, showing results as good as the positive control PDTC (10 μM) (Figure 5).

When analyzing the inhibition of nitric oxide (NO) production in the same LPS-treated RAW 264.7 cells, between 5 and 10 µg/mL, it is possible to see the suppression of NO levels to 80% compared to LPS. This shows similar results to the positive control dexamethasone (DEX). MOL-Flav at 50 µg/mL inhibited NO by around 20% compared to LPS, showing better results than DEX (Figure 5C). Therefore, this finding demonstrates that 50 µg/mL is the more effective concentration and MOL-Flav potently inhibits the NF-kB pathway with a consequential downregulation of inflammatory mediators.

## 4. Discussion

Products of natural origin may vary in composition due to agricultural conditions [1], which can affect the standardization of raw materials from herbal medicine and the quality control of the final product. However, few studies emphasize the difference in polyphenol composition relative to soil cultivation condition, harvest season, or extraction used [17,21,22], or a better alternative, facilitating the quality control and industry scaling up. *Moringa oleifera* is globally recognized for its nutritional and pharmacological properties, correlated to the high content of flavonoids in its leaves. These flavonoids are relevant as an antioxidant and anti-inflammatory agent and may have many potential health applications due to their action on different targets [1,2,6]. Thus, this study presented the optimization process from cultivation conditions to the optimized hydroethanolic extraction of flavonoid concentrate from *M. oleifera* leaves. We also performed a chemical profile characterization by mass spectrometry and evaluated anti-inflammatory properties using an in vitro model. To the best of our knowledge, this is the first study aiming to reveal the correlations from cultivation until obtaining the dried extract (input) with quality control of the phytochemical profile of *M. oleifera* leaves in Brazil.

The most significant advantages of ultrasound-assisted extraction yield of flavonoids compared with traditional techniques are lower extraction temperature, reduced extraction time, lower solvent required, increased yield, compatibility with green-extraction principles, and lower operating costs [18,39]. Our optimized method found that 80% ethanol was the best extraction solvent for flavonoids, which agrees with the literature [1]. Our findings for TPC showed that 50% ethanol was able to extract more phenol content (Figure 2). In the same manner, Rodriguez-Perez et al. [39] found that ultrasound extraction percentage significantly affects phenolic extraction, and 50% ethanol was the best condition. Putting together our results, we can maintain the pharmacological properties of this raw material according to its quality control through the chromatographic profile and TPC.

The literature reports the frequent use of organic solvents such as chloroform-methanol-water (2:1:1) for metabolomics studies [26] or methanol-water for polyphenolic extraction from *M. oleifera* leaves [22]. However, herein, we are focused on future food or pharmacological applications of *M. oleifera* extracts, so the hydroethanolic solvent has been selected. Ethanol is a biocompatible solvent and can extract polyphenols due to the different polarities of the bioactive phytochemical constituents [1]. It is an excellent choice to standardize the process to obtain a raw material that may be commercialized in the future to develop new foods, pharmaceutical products, or cosmetics [40].

The content of secondary compounds is influenced by location, as different cultivation locations may interfere with the chemical profile of *M. oleifera* leaves and may lead to varying nutritional or pharmacological profiles [8,14,22]. Studies have evaluated the influence of edaphoclimatic factors on the variation in bioactivity and total phenolic compounds from *M. oleifera* leaf extracts [21,23]. This study found the most significant yield in MIN-C soil fertilizer, and the optimized long-term yield was found in MIN-B condition (Table 3). This long-term yield can be explained based on a study by Rufai et al. [42], which showed that BioChar in soil can significantly affect the plant height, root number percentage, and root length, which can be responsible for the long-term effect. It is possible to notice that the yield content does not reflect the bioactive compound content.

Focusing on the future use of MOL-Flav as a portion of food or pharmaceutical products, we seek to optimize the extract for the highest content of flavonoids, phenols, and proteins and the lowest sugar content. This was achieved in the MIN cultivation condition and the first harvest in the drought season (Figure 3). These results are challenging to discuss since similar studies in the different aspects analyzed are scarce, so we can explore them in parts. The higher TPC and TFC in the drought season corroborate with the results found in the study compared with the rainy season, Vázquez-León et al. [17]. They concluded that when there is an increase in the solar radiation level, such as UV, the production of total phenolic compounds is positively affected in *M. oleifera*.

The control condition (untreated soil cultivation) presents a high TPC and TFC, which may be related to soil type (sandy texture). These results agree with those obtained by Chludil et al. [43], who identified a high content of flavonoids in Quinoa (*Chenopodium album*) grown in deteriorated, sandy soils that have the characteristic of not properly retaining nutrients, allowing water leach. The literature described that *M. oleifera* is adapted to a wide variety of soil types but prefers arid and sandy soil conditions [12], which were the conditions present in the cultivation of this study in Northeastern Brazil.

Previous *M. oleifera* harvest studies have assessed the influence of the type of soil fertilization [8,23,44], harvest location [21], and different cultivars [46,47] on the quality of the leaves produced. It is essential to state the cultivation conditions because many variables are present when comparing open-field cultivation to greenhouse environments [1]. To our knowledge, only two studies have been performed with Brazilian samples until now, but they focused on the bioguided extraction of phenolic compounds [37] and their antihyperglycemic activity [38]. This is the first analysis of the species under specific cultivation conditions, and the influence of these different aspects is not fully understood. Thus, more studies characterizing this species grown in Brazil are needed.

The MOL-Flav was characterized by HPLC-ESI-QTRAP-MS/MS and showed the prevalence of flavonoids on the fingerprint (Figure 4), according to the literature [3,22]. Accordingly, MOL-Flav analyses revealed the presence of nine majority compounds: quinic acid (compound **1**), which has been previously identified in *M. oleifera* leaves [35,39]. Compound **2** was described, but it remains unidentified because it was not possible to elucidate its structure. *C*-glycosidic flavonoids were identified as apigenin-6,8-*C*-dihexose (vicenin-2, compound **3**), which were already previously identified in *M. oleifera* [40]. Compounds **4** and **5** were characterized as isomers of apigenin hexoside (vitexin and isovitexin), and they were previously identified in *M. oleifera* extracts by Rodríguez-Pérez et al. [39].

However, it was not possible to identify the type of sugar linked *O*-glycosidic flavonoids, so the compounds were characterized as described in Table 4. Quercetin-3-*O*-hexose (compound **6**) was already described for *M. oleifera* leaves [21]. Quercetin-3-*O*-acetyl-hexose (compound **7**) was previously identified in *M. oleifera* leaves [37]; kaempferol-3-*O*-hexose (compound **8**) [44]; kaempferol-3-*O*-acetyl-hexose (compound **9**). However, complementary analyses need to be performed in order to characterize the type of hexose that is fully linked. The hexose most reported for the *M. oleifera* is glucose, such as kaempferol-3-*O*-glucoside, as described by Xu et al. [3]. Furthermore, quantitative experiments such as metabolomics should be performed to evaluate more plant samples and the relationships between the parameters of temperature, season, and type of soil fertilization to better identify the related variables.

The antioxidant and anti-inflammatory in vitro activities of MOL-Flav were analyzed in LPS-treated Raw 264.7 cells. MOL-Flav was not cytotoxic at 5, 10, and 50 µg/mL. However, at 125 and 250 µg/mL, cell viability was significantly reduced compared to the control group (Figure 5). Similarly, *M. oleifera* extracts did not have cytotoxicity in HepG2 cells [12]. Moreover, we found that MOL-Flav is a robust free radical scavenger and can be considered a good source of natural antioxidants to treat inflammation-associated diseases (Figure 5). Previously, Esposito et al. [34] showed that phenolic compounds block the inflammatory process by inhibiting ROS formation, thereby reducing the formation of pro-inflammatory cytokines. The pathway of anti-inflammatory activity of MOL-Flav needs further investigation. NO is particularly associated with chronic inflammation [34,35] but was mainly reduced by MOL-Flav at 50 µg/mL (Figure 5). This activity may be related to the concentrated content of flavonoids.

As far as we know, we have evaluated the influence of different cultivation and harvests seasons of *M. oleifera* leaves on the content of specialized metabolites by open-field cultivation in Northeastern Brazil for the first time. This standardized process can be considered a simple method easily transferred to industrial scale in the pharmaceutical and food industries for producing raw material from *M. oleifera* leaves. Thus, this study may assist in planning strategies for quality control, can be used to select the best conditions for agricultural cultivation, and to improve the raw material from *M. oleifera* leaves, in addition to serving as a reference for its future application in Brazil.

## 5. Conclusions

This study demonstrated a successful procedure to enhance the flavonoid content from *M. oleifera* leaves from the standardized soil cultivation condition, harvest season, to the developed and optimized ultrasound extraction. The agricultural factors may significantly impact the secondary metabolite contents in *M. oleifera* leaves. For the evaluated parameters, the mineral cultivation and drought season conditions enhance the flavonoid, phenolic, and protein contents in *M. oleifera* leaves. The MOL-Flav extraction process carried out with 80% ethanol is in line with the global trends in the development of food technology. Phytochemical analyses characterized nine major compounds by HPLC-QTRAP-MS/MS, including six glycosylated flavonoids derived from apigenin, quercetin, and kaempferol. The anti-inflammatory properties of MOL-Flav were suggested using an in vitro model of acute inflammation in LPS-treated RAW 264.7 cells. MOL-Flav displays no cytotoxicity and promising anti-inflammatory activity. The leaves grow fast, which enables their commercial production at a large scale. Overall, this study contributed to the development of a flavonoid-rich extract from *M. oleifera* leaves cultivated in Brazil, since cultivation until the product (dried extract with defined flavonoid content). It may be used as a reference for its future application in at input, food, and pharmaceutical companies.

## Figures and Tables

**Figure 1 foods-11-01452-f001:**
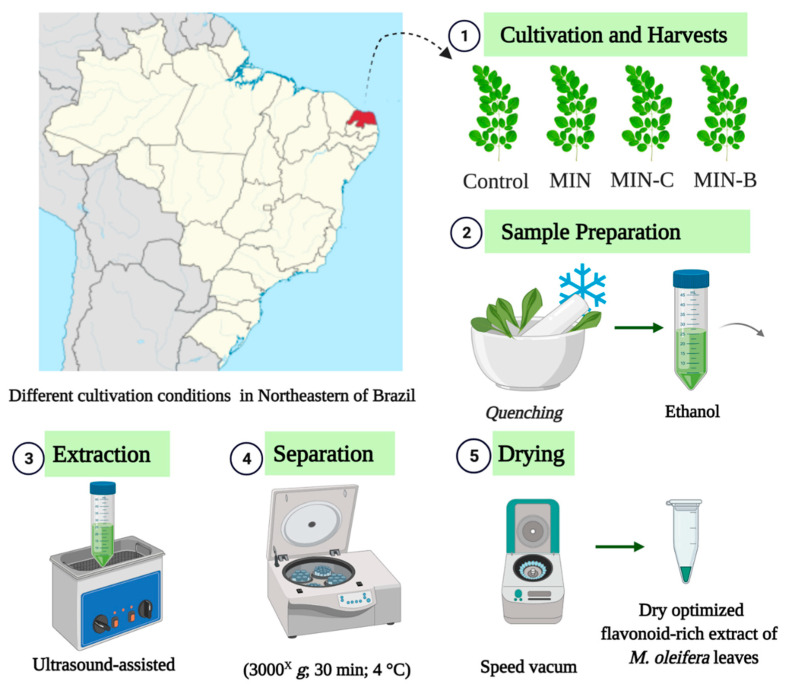
Graphical presentation of the protocol to obtain an optimized process of flavonoid-rich extraction from *Moringa oleifera* leaves employing ultrasound-assisted extraction (UAE). (**1**) Cultivation conditions: Control (untreated local soil); MIN (100% mineral cultivation); MIN-C (50% mineral soil fertilizer with 50% composting); and MIN-B (50% mineral cultivation with 50% Biochar (3.0 t/ha); (**2**) Sample preparation with metabolic *quenching* and ethanol as extraction solvent; (**3**) Ultrasound-assisted extraction; (**4**) Supernatant separation by centrifuge; (**5**) Sample dryer and the final optimized flavonoid-rich extract of *M. oleifera* leaves.

**Figure 2 foods-11-01452-f002:**
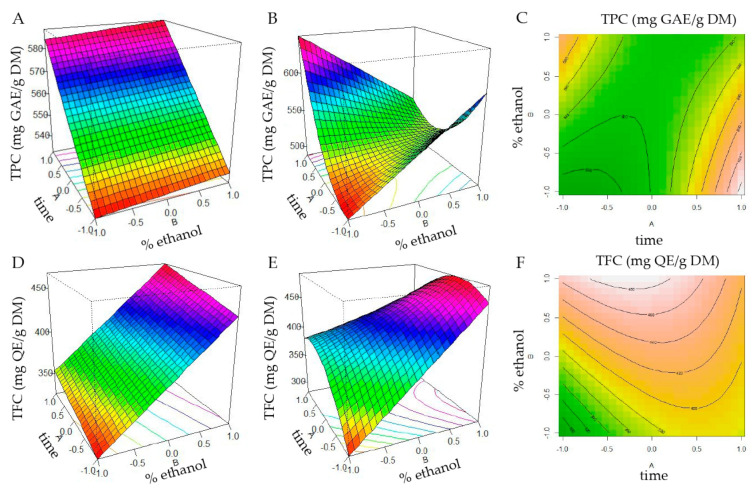
Response surface and contour plots showing the interactive effects of extraction independent variables on the total phenolic content (TPC) and total flavonoid content (TFC). (**A**) Model fitted to first order model for TPC; (**B**) Model fitted to second-order model for TPC; (**C**) contour plots for TPC; (**D**) Model fitted to first order model for TFC; (**E**) Model adjusted to second-order model for TPC; (**F**) Contour plots for TPC.

**Figure 3 foods-11-01452-f003:**
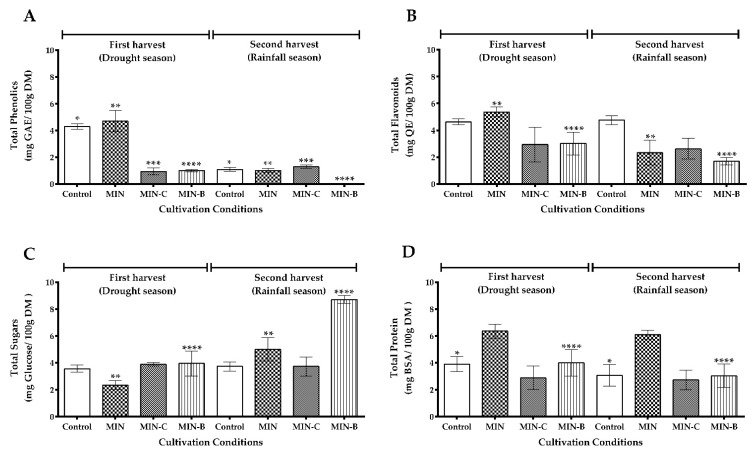
The analysis of flavonoid-rich extract from *Moringa oleifera* leaves (MOL-Flav) under different cultivation conditions and harvests. The total content of (**A**) phenolics; (**B**) flavonoids; (**C**) total sugars; (**D**) proteins. The result was calculated as mean ± SEM (*n* = 3). One-way ANOVA and Tukey’s post hoc test were used to compare the extractions for the total phenolic, flavonoid, sugar, and protein content. * *p* < 0.05, a statistically significant difference for Control between the different harvests. ** *p* < 0.05, a statistically significant difference MIN between the different harvests. *** *p* < 0.05, a statistically significant difference for MIN-C between the different harvests. **** *p* < 0.05, a statistically significant difference of MIN-B between the different harvests.

**Figure 4 foods-11-01452-f004:**
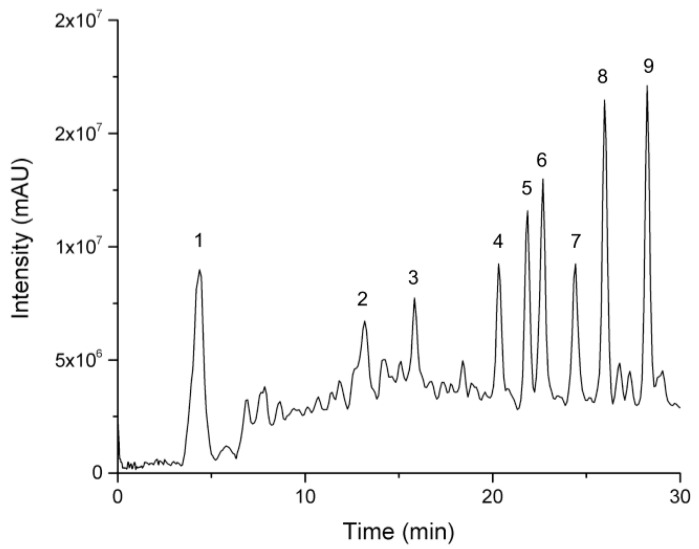
Fingerprinting by ion total chromatogram (TIC) of flavonoid-rich *M. oleifera* leaf extract (MOL-Flav) in the mineral cultivation condition (MIN) first harvest by HPLC-ESI-QTRAP-MS/MS analysis in the negative ion mode. The majority of compounds are identified as numbers (1–9) and explained in the Table 4.

**Figure 5 foods-11-01452-f005:**
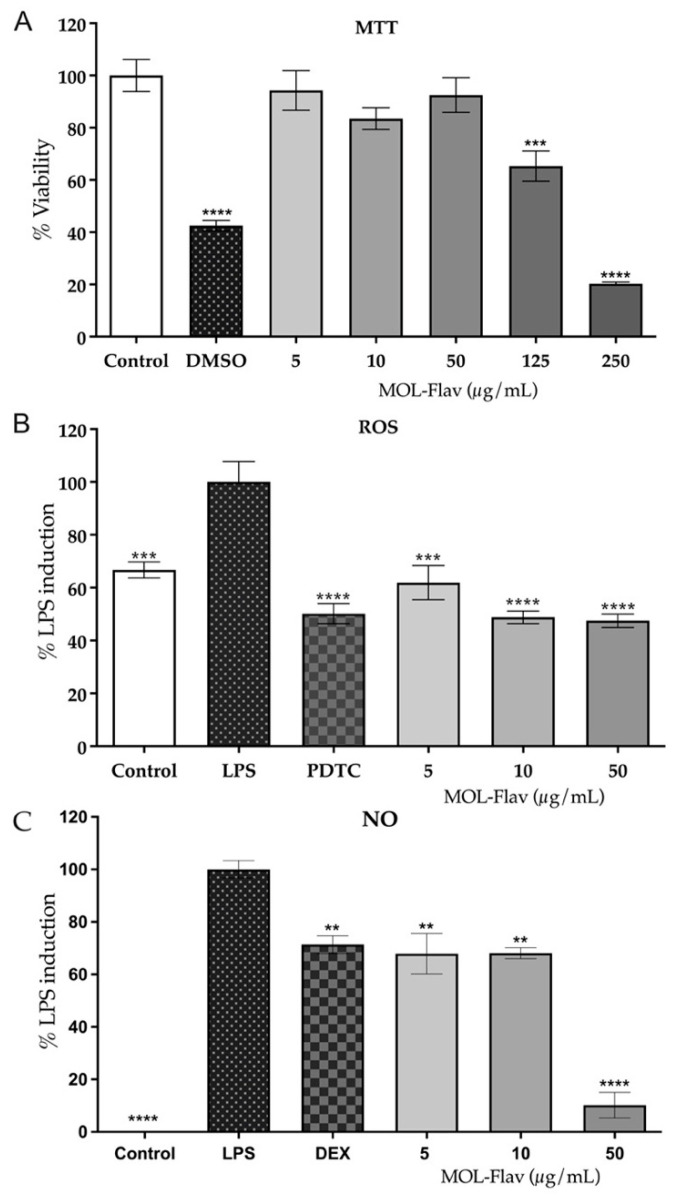
Cell viability (MTT) and the effect on intracellular reactive radical species (ROS) and nitric oxide (NO) of flavonoid-rich extract from *M. oleifera* leaves (MOL-Flav) on murine macrophage RAW 264.7 cells. (**A**) Inhibitory effects of MOL-Flav on RAW 264.7 cells, Control (80% ethanol), DMSO (1% dimethyl sulfoxide); (**B**) ROS production, Control (80% ethanol), LPS (100 μg/mL, lipopolysaccharide), PDTC (10 μM, pyrrolidinedithiocarbamate ammonium); (**C**) NO production. Control (80% ethanol), LPS (1 μg/mL), DEX (10 μM dexamethasone). Data represent the mean ± SEM from three independent experiments. One-way ANOVA was followed by the post hoc Dunnett test. *** *p* < 0.001, **** *p* < 0.0001 vs. Control group (MTT). ** *p* < 0.01, *** *p* < 0.001, **** *p* < 0.0001 vs. LPS-treated cells (ROS and NO).

**Table 1 foods-11-01452-t001:** Design and results for total phenol content and total flavonoid content of *M. oleifera* leaf extracts obtained by preliminary ultrasound-assisted extractions (UAE) experimental design.

Experiment	Factors		Responses	
	Extraction Time (min)	Ethanol (%)	Total Phenol Content (mg GAE/g DM)	Total Flavonoid Content (mg QE/g DM)
1	10 (−1)	50 (−1)	496.48	274.69
2	10 (−1)	50 (−1)	447.63	281.65
3	10 (−1)	50 (−1)	518.42	288.17
4	20 (1)	50 (−1)	663.42	366.89
5	20 (1)	50 (−1)	596.58	379.04
6	20 (1)	50 (−1)	679.21	395.56
7	10 (−1)	80 (1)	635.00	440.35
8	10 (−1)	80 (1)	574,47	478.61
9	10 (−1)	80 (1)	587.10	474.26
10	20 (1)	80 (1)	587.89	401.22
11	20 (1)	80 (1)	529.21	433.39
12	20 (1)	80 (1)	514.47	432.96
13	15 (0)	65 (0)	519.21	405.13
14	15 (0)	65 (0)	497.63	439.48
15	15 (0)	65 (0)	554.74	444.26

Individual data from triplicate analyses. The mg GAE/g DM: gallic acid equivalent/ dry matter. The mg QE/g DM: quercetin equivalent/dry matter.

**Table 2 foods-11-01452-t002:** Analysis of response surface data using multiple linear regression (MLR) and second-order (SO) for the total phenolic content (TPC) and total flavonoid content (TFC).

Factors	Models	*R* ^2^	Residual Standard Error	*p*-Value	*F*-Statistic
TPC	MLR	0.1345	1319.6	0.4203	0.9325
	SO	0.7822	40.59	0.003197	8.589
TFC	MLR	0.6456	297.5	0.001982	10.93
	SO	0.9524	17.25	0.0000014	50.07

**Table 3 foods-11-01452-t003:** The total yield of *M. oleifera* leaf extracts from different soil cultivation and different harvest seasons.

Cultivation Conditions	1st Harvest August 2018 (Drought Season)	2nd Harvest February 2019 (Rainfall Season)
Untreated soil cultivation (Control)	5.97%	4.99%
Mineral cultivation (MIN)	6.37%	5.46%
Mineral cultivation with composting (MIN-C)	7.29%	5.79%
Mineral cultivation with Biochar (MIN-B)	6.40%	6.15%

Average data from triplicate analyses.

**Table 4 foods-11-01452-t004:** HPLC-ESI-QTRAP-MS/MS data of the compounds identified in *M. oleifera* hydroethanolic leaf extract (MOL-Flav) in the first harvest of the mineral cultivation condition (MIN).

Nº	RT (min)	MS (*m*/*z*)	MS/MS Fragments (*m*/*z*)	Molecular Formula	Tentative Assignment
**1**	4.33	191.02 [M-H]^−^	191.04; 173.08; 126.86; 92.78; 86.79; 84.79	C_7_H_12_O_6_	quinic acid
**2**	13.15	611.35 [M-H]^−^	371.26; 269.11; 239.18; 209.12; 167.00	-	unidentified
**3**	15.84	593.23 [M-H]^−^	397.38; 383.29; 353.25; 325.25	C_27_H_30_O_15_	apigenin–6,8–*C*–dihexose (vicenin–2)
		595.40 [M+H]^+^	379.25; 349.16; 325.27; 307.18; 295.20		
**4**	20.33	431.33 [M-H]^−^	341.28; 311.23; 283.21	C_21_H_20_O_10_	apigenin hexose isomer I (vitexin)
		433.39 [M+H]^+^	397.17; 379.29; 367.32; 337.38; 313.28; 284.26; 283.29		
**5**	21.85	431.27 [M-H]^−^	341.26; 323.24; 311.20; 283.24; 269.30	C_21_H_20_O_10_	apigenin hexose isomer II (isovitexin)
		433.39 [M+H]^+^	397.22; 367.28; 337.23; 313.19; 283.24		
**6**	22.68	463.33 [M-H]^−^	301.16; 300.21; 271.18; 255.24; 179.09; 150.98	C_12_H_20_O_12_	quercetin–3–*O*–hexose
		303.35 [M+H]^+^	304.49; 303.12; 229.24		
**7**	24.40	505.27 [M-H]^−^	301.18; 300.19; 271.20; 255.13; 179.08; 150.92	C_23_H_21_O_13_	quercetin–3–*O*–acetyl-hexose
		303.47 [M+H]^+^	304.51; 303.18; 136.92		
**8**	25.97	447.22 [M-H]^−^	327.29; 285.18; 284.21; 255.18; 255.18; 227.20; 179.13; 151.06	C_21_H_20_O_11_	kaempferol–3–*O*–hexose
		287.31 [M+H]^+^	287.12; 213.35; 165.18; 153.21		
**9**	28.24	489.22 [M-H]^−^	286.13; 285.15; 284.20; 255.23; 229.20; 162.93	C_23_H_22_O_11_	kaempferol–3–*O*–acetyl-hexose
		287.43 [M+H]^+^	287.14; 213.23; 153.08		

## Data Availability

We did not include such data in this study.

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
