# Peer review of "The First Optimization Process from Cultivation to Flavonoid-Rich Extract from Moringa oleifera Lam. Leaves in Brazil"

_foods, 2022, doi:10.3390/foods11101452_

Round 1
Reviewer 1 Report
My recommendations for the authors of study are,
Improve the discussion of biomedical results as they use DEX and PDTC compounds as controls to compare the ability of extract both to inhibit iNOS and neutralize ROS, respectively, but the effect was partial. It is also interesting that they use the Griess assay to measure (nitrites and nitrates such as possible products of oxidation of nitric oxide and do not have a basal reading of the control or is minimal. Please explain.
Also, I would like to know why it was not interesting to use my papers as part of your study.
Erick Sierra-Campos, Monica A. Valdez-Solana, et al., Standarization based on chemical markers of moringa oleifera herbal products using bioautography assay, thin layer chromatography and high performance liquid chromatography-diode array detector. Malaysian Journal of Analytical Sciences Volume 24, Issue 3, 2020, Pages 449-463 https://mjas.analis.com.my/mjas/v24_n3/v24_n3.html
I hope my comments contribute to the quality of the manuscript.
Author Response
Q1. Improve the discussion of biomedical results as they use DEX and PDTC compounds as controls to compare the ability of extract both to inhibit iNOS and neutralize ROS, respectively, but the effect was partial. It is also interesting that they use the Griess assay to measure (nitrites and nitrates) such as possible products of oxidation of nitric oxide and do not have a basal reading of the control or is minimal. Please explain.
To betther understanding the use of DEX and PDTC as positive standards, we improve the discussion with the inclusion of the following paragraphs:
Page 13:
PDTC (pyrrolidinedithiocarbamat ammonium) is a potent inhibitor of the nuclear factor-κB (NF-κB) signaling pathway, since downregulation of this pathway is essential to attenuate chronic inflammation processes in inflammatory diseases [46], and so the effect of MOL-Flav were analyzed in comparison with PDTC. The MOL-Flav suppressed the generation of ROS in LPS-treated murine macrophage RAW 264.7 cells, between 5 and 50 µg/mL (p < 0.05). MOL-Flav at 10 and 50 µg/mL inhibited the ROS by around 60% com-pared to LPS, showing results as good as the positive control PDTC (10 μM) (Figure 5).
When analyzing the inhibition of nitric oxide (NO) production in the same LPS-treated RAW 264.7 cells, between 5 and 10 µg/mL, it is possible to see the suppression of NO levels to 80% compared to LPS. This shows similar results to the positive control dexamethasone (DEX). MOL-Flav at 50 µg/mL inhibited NO by around 20% compared to LPS, showing better results than DEX (Figure 5C). Therefore, this finding demonstrates that 50 µg/mL is the more effective concentration and MOL-Flav potently inhibits the NF-kB pathway with a consequentiale downregulation of inflammatory mediators.
Q2. Also, I would like to know why it was not interesting to use my papers as part of your study.
Thanks for the suggestion. The paper is very interesting and it was cited in our study two times (introduction and discussion-reference 25).
Erick Sierra-Campos, Monica A. Valdez-Solana, et al., Standarization based on chemical markers of moringa oleifera herbal products using bioautography assay, thin layer chromatography and high performance liquid chromatography-diode array detector. Malaysian Journal of Analytical Sciences Volume 24, Issue 3, 2020, Pages 449-463 https://mjas.analis.com.my/mjas/v24_n3/v24_n3.html
Reviewer 2 Report
The authors decided to standardize the best cultivation, harvest season in Brazil, and extraction process conditions to improve and ensure the quality of the final product. The next step were used by a three-level factorial planning and RSM method for optimizes the flavonoids content by extraction process. Ethanol solution (1:10 extraction ratio w/v) was used as a solvent for the extraction. In addition to the concentration of ethanol in solution (50, 65, 80%), the extraction time (10, 15, 20 min) was also considered. The whole experiment was prepared and performed properly and met all the requirements of scientific experiment. It is also worth emphasizing that advanced testing methods, such as HPLC-ESI-QTRAP-MS/MS, used in the experiment, make it possible to measure the concentration of many substances simultaneously in a short time, even in a complex matrix, while ensuring very low limits of quantification. By designing the experiment in this way, the authors were able to demonstrate successful procedure to enhance the content of flavonoids from M. oleifera leaves, once the standardize of the soil cultivation condition, harvest season, to the developed and optimized ultrasound extraction. The authors showed that the most key factors were for cultivation conditions: the mineral cultivation and drought, season conditions enhance the flavonoid, phenolic, and protein content in M. oleifera leaves and extraction process should be carried out with 80% ethanol, which is in line with the global trends in the development of food technology. I support this claiming and I consider this research to be of great value, especially taking into account the beneficial influence of flavonoids on human health. Moreover, the developed extraction method may be used to obtain other biologically active compounds of similar structure also for pharmaceutical purposes. which in my opinion in very interesting way shows the great practical value of this research. All my remarks are listed below:
- Line 147 “repeated as many times as necessary.” What does that mean?
- Figure 3 is poorly visible-I suggest increasing it
Author Response
Q1. Line 147 “repeated as many times as necessary.” What does that mean?
We agree with you. We excluded this sentence of the text because did not help us to understand the process.
Q2. Figure 3 is poorly visible-I suggest increasing it
This figure was revised and we did a upload of a new figure in high resolution.
Reviewer 3 Report
Dear authors,
All suggestions and comments are included in the manuscript text. Please check and correct your manuscript accordingly.

Author Response
We are grateful for all your revisions/suggestions at the manuscript and they were accepted.